Model sensitivity for the prediction of extreme sea level events at a wide and fast-flowing estuary: the case of the Río de la Plata

Matías G. Dinapoli<sup>(\*)(1,2)</sup>, Claudia G. Simionato<sup>(1,2)</sup> and Diego Moreira<sup>(1,2,3)</sup>

<sup>(1)</sup>Centro de Investigaciones del Mar y la Atmósfera (CIMA/CONICET-UBA)

<sup>(2)</sup> Instituto Franco-Argentino para el Estudio del Clima y sus Impactos (UMI IFAECI/CNRS-CONICET-UBA)

<sup>(3)</sup> Departamento de Ciencias de la Atmósfera y los Océanos, FCEN, Universidad de Buenos Aires, Argentina.

(\*) Correspondingauthor:
Intendente Güiraldes 2160 - Ciudad Universitaria
Pabellón II - 2do. Piso
(C1428EGA) Ciudad Autónoma de Buenos Aires - Argentina
Phone: (+54) 11 4787 2693

(+54) 11 4576 3300/09 Ext. 388

Fax: (+54) 11 4788 3572

Email: matias.dinapoli@cima.fcen.uba.ar

#### Abstract

A parameter sensitivity analysis for a pre-operational 2-D barotropic application of the ROMS\_AGRIF ocean model for the forecast of sea surface height (SSH) and currents at the South-Western South Atlantic Continental Shelf with emphasis in the Río de la Plata Estuary is presented. Particularly, the

- 5 interest is on the simulation of extreme storm surges generated by persistent and strong southeasterly winds (Sudestadas) which produce strong floods. Atmospheric models show deficiencies in the forecast of winds during those events. Therefore, linear and quadratic bottom friction, wind speed and direction, and runoff were considered in the sensitivity analysis. The analysis yields to a hierarchy of the impacts of them on the simulated SSH. The most important, with non-linearity in the model response, is wind speed.
- 10 It is followed by the quadratic bottom friction and the runoff, which responses are more linear, and present a regional dependence. Runoff has a larger impact than friction in the upper estuary, which decreases downstream. Non-linearity of wind speed is mainly due to the parametrization of the stress tensor, whereas the interaction with the runoff is not relevant in spite of the huge discharge of this particular estuary. This information allows an optimal calibration of the model with a minimum number 15
- of simulations.

#### 1. Introduction

- The Río de la Plata (RdP, Fig. 1) is one of the largest estuaries in the world (Shiklomanov, 1998). It is formed by the confluence of the Paraná and Uruguay rivers, which form form the second largest basin of 20 South America, after the Amazon (Meccia et al., 2009). The RdP is located at approximately at 35° S on the eastern coast of southern South America, and has a funnel shape, with a length of approximately 300 km and breadths of 40 km at the upper end and 220 km at its mouth (Meccia et al., 2009). The estuary 10 has а mean depth of only m (Balay, 1961)The RdP has a huge runoff with a mean value of around 22,000 m3 s-1, ranking 5th worldwide in water 25 discharge (Nagy et al., 1997; Jaime et al., 2002, Framiñan et al., 1999). Nevertheless, the system presents large variability associated to El Niño - Southern Oscillation cycles (ENSO, Robertson and Mechoso,
- 1998) with peaks as high as 80,000 m3 s-1and as low as 8,000 m3 s-1 have been recorded (Jaime et al., 2002).
- The RdP impacts the nutrient, sediment, carbon and fresh water budgets of the South Atlantic Ocean 30 (Framiñan et al., 1999; Guerrero et al., 2010, among others), affects the hydrography of the adjacent continental shelf, impacts important coastal fisheries, and influences coastal dynamics up to more than 400 km north on the Brazilian shelf (Campos et al., 1999; Framiñan et al., 1999; Piola et al., 2000).

The RdP is of large social, ecological and economic importance for the countries on its shores, Argentina and Uruguay. The capital cities (Buenos Aires and Montevideo) and the main industrial poles and resorts

35 are located on its margins. The estuary is an area of spawning and nursery for a conglomerate of coastal species (see, for instance Cousseau, 1985; Boschi, 1988; Macchi et al., 1996; Acha et al., 1999; Acha and Macchi, 2000; Jaureguizar et al., 2003b; Jaureguizar et al., 2008). The RdP has several important

navigation channels to reach the northern part of Argentina and Paraguay, which are important for the economy of the countries and demand regular dredging. Finally, the estuary is an important amusement zone and the main source of drinking water for the millions of inhabitants in the region. Being the most developed basin of southern South America, the RdP is strongly impacted by anthropogenic actions.

- The RdP is located in one of the most cyclogenetic regions of the world (approximately 8 cyclones per year), associated to atmospheric waves that move along subtropical latitudes of the South Pacific and South American regions, with higher frequency in the warm seasons (Vera et al., 2002). When cyclones develop over Uruguay, they can originate very strong and persistent southeasterly winds (known as *Sudestadas*), with speeds that can easily exceed 15 m s<sup>-1</sup> (Seluchi, 1995; Seluchi and Saulo, 1996). The
- coincidence of large or even moderately high tides and the large meteorologically induced surges during Sudestadas, has historically caused catastrophic floods in the RdP coasts, threatening and claiming human lives and producing major economic and material damages (D'Onofrio et al., 1999). This phenomenon affects, in particular, Metropolitan Area of Buenos Aires City (AMBA), located on the upper RdP estuary. Balay (1961) defined risk water levels over the Tidal Datum of the RdP at the AMBA in 2.50 m
- for alert, 2.80 m for emergency and 3.20 m for evacuation (Escobar et al., 2004). Since records began in 1905, the maximum water level at Buenos Aires was registered in 1940. Enhanced by strong southeasterly winds, it reached 4.44 m above the Tidal Datum, being the tidal height overcome by 3.18 m. More recently, in 1989 and 1993, extreme floods were also experienced at the city. Water levels reached 4.06 m and 3.95 m above the Tidal Datum, being the tidal heights overcome by 3.25 m and 2.49 m,
- respectively (D'Onofrio et al., 1999). Even though the events are not always so extreme, they are frequent, taking place several times every year. It has been suggested that the flooding is mainly due to combination of tides and surge (D'Onofrio et al., 1999), but the effect of the large runoff that characterizes this estuary has not been fully explored yet.

In above described context, the need of forecast models for sea level height (and eventually other variables) prediction in the region is evident. Currently, in the frame of a collaborative project between the Argentinean Hydrographic Service of the Navy (SHN) and the Center for Atmospheric and Oceanic Research (CIMA/CONICET-UBA) the implementation of such a model for the RdP and the adjacent Continental Shelf is being faced. In this sense, the choice of a forecast numerical model is, naturally, strongly dependent on the exactitude and reliability of its solutions. However, all models are imperfect

- abstractions of Nature. Because the discrete nature of the models, the parameterizations and the inaccuracy of the forcing data, numerical solutions always present errors and uncertainties. The errors in forcing data and the uncertainties on the modeling parameters are not independent each other, but can interact in many ways, eventually driving to numeric solutions that might significantly differ of the observations. In this sense, before adopting a particular model for practical applications, it is necessary to
- determine the sensitivity of model solutions to changes in the inputs. The usual manner to do this is by means of a sensitivity assessment (SA), which investigates the relation between the inputs and the outputs of simulation models (Saltelli et al., 2000). SA allows to know how model's solutions change with the diverse parameterizations, forcings and boundary conditions. In addition, it shows where the model needs

improvements contributing to further model development (Norton, 2015) and allows to optimally assembling a regional model through a reduced number of simulations.

The aim of this paper is to perform a SA of a regional application of ROMS\_AGRIF model (Regional Ocean Modeling System, <u>http://www.romsagrif.org</u>) specially implemented for the Northern Argentinean

- Continental Shelf, with focus in the RdP estuary. Besides analyzing the model sensitivity to potential uncertainties in the model parameters, we consider the effect of the large errors of the relative low resolution of atmospheric models in the area, where the width of the estuary turns the proper estimation of wind speed (and direction) in a challenge. In addition, the large runoff of this particular estuary can potentially interact with the surge; therefore, this aspect is also studied. Finally, the non-linear interaction
- between the diverse inputs of the models is evaluated. These results constitute a basic and necessary input for the implementation of an operational model for the forecast of sea surface height (SSH) and ocean currents in this economically, socially and ecologically important region.

#### 2. Input data and methods

#### 15 2.1. ROMS\_AGRIF regionalization to the RdP and the adjacent shelf

ROMS (Regional Ocean Modeling System, <u>https://www.myroms.org</u>) is an ocean numeric model developed by Shchepetkin and McWilliams (2005). It is programmed to simulate physical, biogeochemical, biooptic, sedimentological and sea ice applications. This model has been implemented in several areas (see, for example, Magaldi et al., 2010; Manson et al., 2011; Kumar et al., 2012) including

- in the Patagonian Continental Shelf (for example, Tonini and Palma, 2008; Tonini and Palma, 2009; Combes and Matano, 2014). In this work, we use the ROMS\_AGRIF (Adaptative Grid Refinement in Fortran, <u>http://www.romsagrif.org</u>) version developed at IRD/INRIA (Institut de Recherche pour le Développement / Institut National de Recherche en Informatique et en Automatique; Debreu et al., 2012), including algorithms from MARS3D (Model at Regional Scale, <u>http://wwz.ifremer.fr/mars3d</u>) and 25
- HYCOM (Hybrid Coordinate Ocean Model, <u>https://hycom.org</u>) ocean models, respectively.

Similarly to other studies in the region (see for instance, Meccia et al., 2009), ROMS\_AGRIF model is applied as a hierarchy of one-way nested, barotropic 2-D models. In this case, the RdP is reached through two domains of different resolution and scale. The lower resolution / largest scale "*Model A*" covers an area spanning from 69 °W to 46 °W and from 59 °S to 26 °S (Fig. 1). Horizontal resolution is set to

- 7.50<sup>7</sup>/5.25<sup>7</sup> in the zonal/meridional direction, what is equivalent to approximately 12 km. Model A is used to provide boundary conditions to a higher resolution / lower scale model of the RdP (*Model B*, Fig. 1). This model spans the region between 58.75 °W and 52.50 °W, and 38.20 °S and 32.60 °S, with horizontal resolutions of 2.5<sup>7</sup>/1.75<sup>7</sup> in the zonal/meridional directions, respectively (approximately 4 km). This horizontal resolution is consistent with the 1/3 reduction criteria from father to child models.
- Tidal forcing is introduced by imposing the elevation and barotropic currents at the open boundaries of Model A by means of a bilinear interpolation routine applied to the TPXO8

(<u>http://volkov.oce.orst.edu/tides/tpxo8\_atlas.html</u>) 2.00' resolution solution. The eight most important diurnal and semidiurnal tidal constituents are included in the simulations:  $M_2$ ,  $S_2$ ,  $N_2$ ,  $K_1$ ,  $O_1$  y  $P_1$ .

Given that global bathymetry data bases, like ETOPO, display unrealistic shallow features over Argentinean Continental Shelf, bathymetries for both Model A and B, were built by combining this last

5 data set with data provided by the SHN for depths shallower than 200 m and come from digitalization of charts (SHN, 1986; 1992; 1993; 1999a and b).

#### 2.2. Morris analysis

- SA aims to establishing, using different analysis or methodologies (i) the relative importance or significance of the different inputs; (ii) the occurrence of combined effects of a set of inputs on the model solutions; (iii) and, more broadly, the effect in the output value of changes in a single input, or in a combination of them (Norton, 2015). The SA of this work was made following the methodology suggested by Morris (1991). Morris method is particularly well suited for a model with significant computational overburden as is the case with ROMS\_AGRIF. Its role is to screen the inputs to find out
- which to include in a more detailed analysis (Ruano et al., 2012). The method ranks the inputs according to their influence on each output and highlights their non-linearity. In this context, the inputs comprise the equations' coefficients, parameters and the properties of the forcings, whereas an output is the value of a variable computed by the model or of any statistic derived from it, such as maximum or mean values. The hypothesis of SA is that every input and output can described by a single number (scalar).
- The Morris method works by stepping k inputs along r trajectories, where  $r \ll k$ . Each trajectory is initialized at a random position within the inputs space hypercube formed by the considered ranges of variability of the diverse inputs. Along of a trajectory, into the hypercube, the inputs are perturbed one at a time, with a fixed step size (at most half the range of the input). The changes in the output due to the r changes in every input are then studied as a sample. A large mean of the absolute values of the changes or
- elementary effects due to a particular input (Campolongo et al., 2007) indicates a large influence of that input on the model solution. A large standard deviation, in turn, indicates that the effect depends strongly on the input values, implying strong non-linearity including multilinearity (Morris, 1991).

In this work, the output function chosen to be evaluated in the SA is the root mean square error (RMSE) with respect to the *in situ* observed hourly sea surface height at Palermo and Oyarvide tidal stations (Fig.

1). The RMSE represents an overall error, which is very much related to correlation. In addition to the SA, the comparison of RMSEs for the set of simulations also gives an approach to the set of input that yields to a model solution that approximates the observed signal, which is helpful for an eventual fine calibration. To allow for the inter-comparison among inputs in spite of their order of magnitude, changes were computed using the normalized derivate (Norton, 2015), according to Eq. (1):

$$35 \qquad \frac{\delta y_{/y}}{\delta p_k/p_k} = \frac{p_k}{y} \frac{\delta y}{\delta p_k} \tag{1}$$

Sudestada.

25

where y is the output and  $p_k$  is one of the k inputs. The reader is referred to Norton (2015) for further details on the methodology.

# 2.3. Uncertainties in the meteorological data for ocean model forecast/hindcast at the region of 5 interest

The RdP and the adjacent Continental Shelf are very sensitive to atmospheric forcing, in particular to surface winds (Simionato et al. 2004a, b, 2005a, b, 2006a, b, 2007; Meccia et al., 2009). Both direct observations and numerical models have shown that the wind driven circulation at the estuary can be explained in terms of two modes of circulation. The first, prevailing for winds with a cross-river
component, is related to an inflow/outflow of water at the exterior part of the RdP. The second mode dominates when the wind blows along the estuary axis and has a very distinctive pattern of significant sea level increase or reduction at the upper part of the estuary. In particular, this last mode accounts for the

Despite its importance, the availability of appropriate data to force an ocean forecast model in the region

can become a major problem. Indeed, the different atmospheric reanalysis databases (which are, in theory, much better than forecasts) constructed combining numerical models and observations show important differences when compared to the scarce direct observations over the water in the region. As an example, Fig. 2 shows the wind stress module and direction from various data sets for a strong Sudestada event occurred in May 2000:

(i) direct observations collected at Pontón Recalada station (near to Montevideo, Fig. 1) as red dots; this data have a temporal resolution of 3 hours, i.e. 8-daily.

(ii) 4-daily reanalysis from the National Center for Environmental Prediction/National Center for
 Atmospheric Research – Reanalysis 2 (NCEP/NCAR RII) (Kanamitsu et al., 2002) as green triangles;

(iii) 4-daily reanalysis from the European Centre for Medium-Range Weather Forecast (ECMWF ERA-INTERIM) (Dee et al., 2002) as blue triangles;

(iv) 4-daily Blended Sea Winds (BSW) (Zhang et al., 2006) developed by the National Oceanic and Atmospheric Administration (NOAA), in which wind speeds are generated by blending observations from multiple satellites, and the wind directions come from two sources depending the reanalysis data (NCEP/NCAR RII) and near-real-time forecast data (ERA-INTERIM), as black diamonds.

Table 1 summarizes the main characteristics of every database together with their spatial and temporal resolution.

It is evident from Fig. 2 that all the databases reasonably well represent the wind direction, but they miss represent the wind stress module. For the two re-analyses, wind stress displays an important temporal shift on the maximum of the storm (associated to the low temporal resolution), that will eventually

produce a temporal error in the simulation of this storm surge if those data were used. BSW data, presumably because it is based on remote observations and the combination of re-analyses, provide a

better representation of the wind stress module, but does not improve the timing of the storm. Unfortunately, the few available direct observations of wind over the water in the region are not enough to perform a complete assessment of the most convenient data set for the region of interest. Therefore, it is clear that in spite of the database chosen for a simulation, the wind will probably become the main

5 source of errors and uncertainties in the ocean model solutions. Consequently, we decided to incorporate this variable in the SA developed in this paper.

#### 2.4. Analyzed inputs

The inputs chosen for the analysis of this paper correspond to the main forces in the energy balance of a
 2-D barotropic model for a mighty estuary: the energy dissipation by bottom friction, the atmospheric forcing and the runoff. More specifically:

15

a.

*Bottom friction*: ROMS\_AGRIF considers both a linear and a quadratic coefficient for bottom friction ( $c_1$  and  $c_D$ , respectively). A mean value of  $2.5 \times 10^{-3}$  for  $c_D$  is generally accepted and has been proved to be reasonable for the region (e.g. Simionato et al., 2006a). Here a fine calibration is made, considering a range of variation between  $2.0 \times 10^{-3}$  and  $3.0 \times 10^{-3}$ .  $c_1$  is not a widely-used parameter. ROMS documentation (<u>http://www.myrom.org</u>) suggests that it should be an order of magnitude lower than  $c_D$ ; thus, to explore its effect on the model solutions we have chosen a broad interval ranging between  $1.5 \times 10^{-4}$  m s<sup>-1</sup> and  $5.0 \times 10^{-4}$  m s<sup>-1</sup>.

- b. Wind stress: due to the large difference found between the different wind databases for the region, the evaluation of the impact of the eventual error in the wind forcing is important. We decided to analyze the impact of both the wind speed (w) and direction (Θ) changes. The changes in winds speed are taken into account utilizing a speed factor (I), such that the perturbed speed (w') is w'= Iw. From the observations of Fig. 2, this scalar was chosen to vary between 0.5 and 1.5. In what regards the direction (Θ), its range of variation was estimated using the RMSE of the diverse wind databases of Fig. 2 with
- respect to the in situ observations at Pontón Recalada station. The obtained value was 18°; so, the considered interval ranged from -18° to 18°. NCEP-NCAR RII was chosen to the simulation because this database has been used in another works in the region (for example, Simionato et al., 2006b).

c. *Runoff:* the simulations incorporate the fresh water discharge of the main tributaries to the RdP, the Uruguay River, and the Paraná River in its two branches (Paraná Guazú-Bravo and Paraná de las

- Palmas). Observations (Jaime et al., 2002) indicate that runoff can vary in an enormous range, from less than 8,000 m<sup>3</sup> s<sup>-1</sup> to more than 80,000 m<sup>3</sup> s<sup>-1</sup>. This input varies in low time frequencies because the runoff of the rivers is regulated by large dams located upstream the RdP, and natural variability is mainly dominated by the inter-annual ENSO cycles. Even though its effect can in principle be regarded as not significant for short time forecast, it has been recently shown that the tide-current interaction is important
- in the RdP estuary and that it contributes to the mean sea level (Luz Clara Tejedor et al., 2014). It was, therefore, decided to include this input in the analysis for a complete SA.

Lateral diffusion (v) is not considered in the SA because it is known that this parameter does not produce significant changes in 2-D barotropic models (e.g., Simionato et al., 2004b; Bastidas et al., 2016) and this was verified in a preliminary analysis not shown in this paper for the region of interest (Dinapoli, 2016). ROMS documentation suggests typical values of this input ranging between 1,000 and 0 m<sup>2</sup> s<sup>-1</sup>. Hence, v

was set at 0 for all the simulations. The chosen ranges of existence for the diverse inputs are summarized in Table 2.

As ROMS model let impose wind stress field ( $\tau$ ) as well as wind field (u), we chose the first one to set the surface boundary condition. The conversion from wind vectors is made using the bulk formula Eq. (2):

$$\tau_i = c_D^W \rho_A w u_i \tag{2}$$

where *i* represents either the zonal (*x*) or the meridional (y) wind component,  $c_D^w$  is the wind drag coefficient,  $\rho_A$  is the air density,  $u_i$  is the wind component and *w* is the wind speed. For the parameterization of the wind drag coefficient we used the classical expression Eq. (3):

$$c_D^w = \begin{cases} 1.1 \times 10^{-3} & , w 

suggested that the inputs that lay to the right of that line have a mean significantly different from zero. The fact that the distribution of the inputs is not same for both stations suggests that inputs are sensitive to local hydrodynamics. However, in both cases wind speed was the most important input of the analyzed set, as illustrated by its largest mean. Furthermore, the speed presents a large variance compared to the

- 5 other inputs, indicating its non-linear effect on the simulation (Morris, 1991). After wind speed; c<sub>D</sub> and Q are the inputs which sensitivity is more important. Whereas c<sub>D</sub> seems to have an impact in the output of the model over the entire RdP, it is overcome by the impact of the changes in Q in the upper estuary. This is because in this last region there is a strong interaction between the tide and the runoff (Luz Clara Tejedor et al., 2014); this last, in turn, is very large in the RdP. This way, the rate of dissipation of kinetic
- 10 energy by bottom friction is not enough to produce a significant attenuation of the signal. Towards the outer estuary currents are significantly reduced due to widening and deepening of the RdP; thus, the dissipative effects become relatively stronger and the impact of c<sub>D</sub> increases compared to that of Q.

Concerning  $\Theta$  y c<sub>1</sub>, even though they produce statistically significant changes, their impact is much lower than those of the other analyzed inputs. Consequently, the selection of c<sub>1</sub> can be regarded as fine calibration.

In spite of the wide range of variability considered, uncertainties in wind direction do not have a major effect on the simulated signal. This can be attributed to the fact that the chosen variability range (in turn related with the observed differences between wind observations and simulations) maintains the wind directional in the range that produces one of the prevailing modes of circulation of the RdP estuary identified by Simionato et al. (2006a, b).

Summarizing, model solutions are highly (and non-linearly) sensitive to uncertainties in wind speed, but less sensitive to uncertainties in wind direction. Concerning dissipation by bottom friction, only the quadratic term produces significant changes in the simulations. Finally, the runoff has a large impact in upper estuary, which decreases downstream. Although its impact is not as large as that of wind speed, it is

25 comparable to the effect of  $c_D$  and, therefore, it is necessary to consider it in the simulations.

Since 204 different simulations were run, it is worthwhile to look for the best combination of inputs that produce the "optimal" solution. The selection was made computing the correlation coefficients, the gradients of the regression line of simulations *vs.* observations, and the RMSE. Whereas the first statistic measures the covariance between observations and simulations with regards to temporal variability, the

- second one is an indicator whether the numerical solution fits or not to the magnitude of the observations (Meccia et al., 2009). The criterion of selection was correlation more than 0.9, gradient between 0.8 and 1.2, and a minimal RMSE. Fig. 4 shows the so obtained optimal signals (solid red lines) compared to observations (dashed blue lines) for Palermo (left panel) and Oyarvide (right panel) stations. The corresponding inputs were  $c_p = 2.20 \times 10^{-3}$ ;  $c_l = 1.50 \times 10^{-4}$  m s<sup>-1</sup>; I = 1.15;  $\Theta = 1.5^{\circ}$ ; Q = 22,400 m<sup>3</sup> s<sup>-1</sup>. The
- values of bottom friction can be utilized as a first estimator for posterior fine calibration. The wind inputs indicate that the direction is well captured by the atmospheric model but that wind speed is underestimated, what is consistent with the conclusions of another works (see, for instance, Simionato et al., 2006a) and with what was observed from Fig. 2. On the other hand, the optimal value for Q

corresponds to that observed during the particular case of Sudestada analyzed, confirming that the inclusion of this variable contributes to an improvement of the simulation.

# 3.2. Quantification of the wind and continental discharge influence

- The Morris analysis discussed in the previous sections showed that it is strongly non-linear. An unanswered question is whether this non-linearity is intrinsic of the input (due to the formulation of the wind stress, see equation 2) or it results of the interactions between the storm surge and the runoff. To further explore this issue, a new set of simulations was run, in which only the effect of I and Q were considered. Taking into account the results of the Morris analysis discussed in the previous section, the
- intervals of variability of the inputs were maintained (Table 2) but the discretization was changed, choosing finer intervals for I (steps of 0.1) and coarser intervals for Q (steps of 14,000 m<sup>3</sup> s<sup>-1</sup>). The other inputs were fixed as  $c_D = 2.20 \times 10^{-3}$ ;  $c_1 = 1.50 \times 10^{-4}$  m s<sup>-1</sup> and  $\Theta = 0^{\circ}$ .

RMSE between the observed and simulated signals was computed once more with the aims of (i) to analyze an eventual interaction between wind speed and runoff; and (ii) to use the results to derive the set

- of inputs that drives to the minimum error, in order to make a second approach to the calibration of the inputs. Fig. 5 shows contour lines of RMSE as a function of I and Q for Palermo (left panel) and Oyarvide (right panel) stations. For Palermo, a sector with a minimum RMSE is observed around I = 1.1 and Q = 22,000 m<sup>3</sup> s<sup>-1</sup>; this means that the numerical solution is optimal when discharge is fixed around the observed value, but wind speed must be augmented for about 10%. For Oyarvide station, results
- indicate that solutions are completely insensitive to changes in Q, and much less sensitive to changes in wind speed than Palermo. Results are also consistent with the empirical factor of correction for wind speed proposed by Simionato et al., (2006b) for NCEP/NCAR RII in the region.

The pronounced slope of the RMSE isolines in the direction of the runoff axis for Palermo, suggests that the non-linear response of the model is mostly due to the intrinsic non-linearity of wind speed, whereas

runoff plays a secondary role.

# 4. Summary of conclusions and final remarks

- In this work, we discuss a sensitivity analysis based on Morris methodology, which is particularly well suited for models with large computational demand, to determine the sensitivity of numerical solutions for the Southwestern Atlantic Continental Shelf with emphasis in the wide and fast flowing Río de la Plata Estuary to different parameters. An evaluation of the overall model performance during an intense storm surge was conducted, which permitted the evaluation of the overall model precision and accuracy during the most critic events for the inhabitants of the region and navigation, known as Sudestadas. The
- results from the sensitivity analysis reduce the required number of simulations needed to calibrate the model, which is expected to be implemented in an operational ocean forecast system for the region.