# Peer review of "Model sensitivity for the prediction of extreme sea level events at a wide and fast-flowing estuary: the case of the Río de la Plata"

_Natural Hazards and Earth System Sciences, 2016_

## Referee Comment (RC1) · Anonymous Referee #1 · 10 May 2017

This paper by Dinapoli et al. presents a sensitivity analysis of extreme water level predictions at the Rio de la Plata Estuary. This topic fits well with NHESS, but, in the present form, I cannot recommend its publication, for the main following reasons:

-Once optimized, the model predictive skills are still not convincing, at least based on figure 4. Indeed, there might be an inconsistency between this figure, where the model departs from the observations by 0.5-1m a large part of the time, and figure 3 and 5, which suggests errors < 0.5 m. Considering that the correct figure is figure 4, I would rather expect ERMS of the order of ∼0.3-0.4 m, which is not convincing compared to recent studies (e.g. Bunya et al., 2010 ; Bertin et al., 2015). A possible reason for these relatively weak predictive skills is the wind that was selected (NCEP), which is

way too coarse to compute storm surges accurately. From my own experience, a wind forcing of at least 0.2° resolution should be employed. Why not considering the CFSR reanalysis (Saha et al., 2010), which is fully available and offers a spatial resolution of 0.2° and a time resolution of 1 h.

-The authors evaluated directly total SSH, which does not allow to understand if their errors come from tides, storm surges or a bit of both. What is the impact of baroclinic circulation in such a large estuary, baroclinic effects are even not mentioned in the paper? I would suggest separating model results for tide only, surge only and model + surge. This would namely improve understanding, tide-surge interactions (e.g. Idier et al., 2012), which might be quite important in such a large estuary. I understand that it would lengthen the paper substantially but the present paper is quite short compared to the average of NHESS papers.

-There is a confusion between wind speed and surface stress. Why evaluating surface stress and not wind speed directly? Indeed, using the surface stress implies defining a drag coefficient, which is a very large source of uncertainty, namely because it depends on wind speed but also, for a given wind speed, on the sea-state. In the end of section 2.4, the author state that "the wind drag coefficient has a relative small effect on the parameterization...": what does it means given that $Cd$ increases linearly with the wind speed? What is the idea behind keeping $Cd$ constant while it is known that it increases with wind speed? Considering the sensitivity analysis on the wind speed where I ranges from 0.5 to 0.5, eq. (3) implies that $Cd$ should vary by a factor of 3.

If the authors are willing to revise their manuscript or to submit it to another journal, I have also identified the following along-the-text less important problems:

-P2, L12: "surface stress" is more common.

-P2, L18-23: strange to start the introduction by presenting the study area.

-P2, L25: what do you mean by "system", has ENSO broader impacts than river runoff?

-P2, L35: "along its margins"

-P3, L5: I strongly doubt that the RdP is located in one of the most cyclogenetic region of the world. Regarding tropical storms, the Southern Atlantic is on the contrary the only Ocean where the cyclonic activity is almost nil (https://earthobservatory.nasa.gov/IOTD/view.php?id=7079). Regarding extratropical storms, please consider that, in winter, storms develop continuously at high latitudes.

-P3, L8: "yields" rather than "originate".

-P3, L16-20: please explicit the corresponding surges.

-P3, L30: a forecast is never "exact", rather use "quality".

-P4, L7: explain why a wide estuary turns more challenging the estimation of the wind speed and direction.

-P4, section 2.1: rather explain what are the key differences of ROMS_AGRIF compared to the original version. As said above, please better justify why baroclinic circulation can be neglected.

-P5, L9: "aims to establish".

-P5, L30: this is not true that RMSE is "very much related" to the correlation. For instance, let's consider a perfect model (RMSE = 0 m) that you divide by a factor of 2: the correlation coefficient will be stick to 1 while the RMSE will get large.

-P6, L6-7: what is the specificity of the RdP compared to any other coastal zone bordered with a large shelf/shallow waters?

-P6, L17: what is the reference height for this open water wind measurements? Usually, this is hardly 10 m and therefore wind speed cannot be compared directly with 10 m wind speed from atmospheric models. If a correction is made for the comparison for instance applying a logarithmic model, which Z0 is used?

-P7, L9: rather "momentum balance".

-P7, L12-18: why combining a linear with a quadratic bottom stress? This is not a common procedure. Please better justify through adequate references.

-P7, section "c": if runoff is that important, why baroclinic circulation is not? Also, better justify through references why tide-surge interactions are huge?

-P8, eq. (3): where this parameterization comes from? Smith and Banke? Large and Pond?

-P10, L5: what does "it" stands for?

Cited references:

Bertin, X., Li, K., Roland, A., Bidlot, J.R. 2015. The contribution of short-waves in storm surges: Two case studies in the Bay of Biscay. Continental Shelf Research 96, 1-15. Bunya, S., Dietrich, J.C., Westerink, J.J., Ebersole, B.A., Smith, J.M., Atkinson, J.H., Jensen, R., Resio, D.T., Luettich, R.A., Dawson, C., Cardone, V.J., Cox, A.T., Powell, M.D., Westerink, H.J., and Roberts, H.J. 2010. A high-resolution coupled riverine flow, tide, wind, wind wave, and storm surge model for Southern Louisiana and Mississippi. Part I: Model Development and Validation. Monthly Weather Reviews 138, 345-377.

Idier, D., Dumas, F., Muller, H., 2012. Tide-surge interaction in the English Channel. Natural Hazards and Earth System Science 12, 3709-3718.

Saha, S., Moorthi, S., Pan, H.-L., Wu, X., Wang, J., Nadiga, S., Tripp, P., Kistler, R., Woollen, J., Behringer, D., Liu, H., Stokes, D., Grumbine, R., Gayno, G., Wang, J., Hou, Y.-T., Chuang, H.-Y., Juang, H.-M. H., Sela, J., Iredell, M., Treadon, R., Kleist, D., Van Delst, P., Keyser, D., Derber, J., Ek, M., Meng, J., Wei, H., Yang, R., Lord, S., Van Den Dool, H., Kumar, A., Wang, W., Long, C., Chelliah, M., Xue, Y., Huang, B., Schemm, J.-K., Ebisuzaki, W., Lin, R., Xie, P., Chen, M., Zhou,S., Higgins, W., Zou, C.-Z., Liu, Q., Chen, Y., Han, Y., Cucurull, L., Reynolds, R. W., Rutledge, G. and Goldberg, M., 2010. The NCEP Climate Forecast System Reanalysis. Bulletin of the American

Meteorological Society 91(8), 1015–1057.

---

## Referee Comment (RC2) · Anonymous Referee #2 · 4 Jul 2017

This is a manuscript which fits the theme of NHESS but which needs major revisions before publication.

The writing can be improved. Some sections can be shortened or summarized. Sometimes (too) many references are attributed to a simple sentence. An example, from the Introduction: "The RdP has a huge runoff with a mean value of around 22,000 m3 s-1, ranking 5th worldwide in water discharge (Nagy et al., 1997; Jaime et al., 2002, Framiñan et al., 1999)." I wouldn't use the terms "huge", nor "runoff". Also don't see the relevance of worldwide ranking. Also don't see why such a minor sentence deserved 4 citations.

[Figure]

The authors reach a somewhat trivial set of conclusions, in that *storm* surge results are most sensitive to the wind forcing and to the bottom roughness – what else should be expected in such a shallow, wide estuary?

A few other remarks:

- Would improve if one or more figures showed the model grids used;

- Why use an ocean model to simulate an estuary of 10m average depth? More flexibility would yield finer resolutions;

- The discussion & conclusions is missing a more comprehensive comparison against similarly-minded papers, e.g. Zijl et al. (2015) where RMSE's are much smaller;

- The discussion & conclusions would benefit from a clear separation of tidal from other mechanisms contributing to the total water levels.

Reference

Firmijn Zijl, Julius Sumihar, Martin Verlaan "Application of data assimilation for improved operational water level forecasting on the northwest European shelf and North Sea" Ocean Dynamics 65(12); November 2015; DOI: 10.1007/s10236-015-0898-7

---

## Author Comment (AC1) · 15 Aug 2017

ANSWER TO THE EDITORS AND REVIEWERS Ref.: Nat. Hazards Earth Syst. Sci. Discuss., doi:10.5194/nhess-2016-393-RC1, 2017 "Model sensitivity for the prediction of extreme sea level events at a wide and fast-flowing estuary: the case of the Río de la Plata" by Matías G. Dinapoli et al.

Dear Dr. Ulbrich, We are grateful to you for giving us the possibility of reviewing the manuscript, and to the reviewers because of their careful and constructive revision, their comments and suggestions, which certainly contributed to a significant improvement of the article. The reviewers' comments were carefully considered and changes

were done accordingly. A discussion of them is given below. We hope you will find the attached revised discussion version of the manuscript suitable for publication in Natural Hazards and Earth System Sciences. Sincerely yours,

Matías G. Dinapoli

Answer to Reviewer 1 comments:

As we mentioned to the Editor, we are grateful for your careful and constructive revision, your comments and suggestions, which certainly contributed to a significant improvement of the article. Your comments were carefully considered and changes were done accordingly. A discussion of them is given below.

This paper by Dinapoli et al. presents a sensitivity analysis of extreme water level predictions at the Rio de la Plata Estuary. This topic fits well with NHESS, but, in the present form, I cannot recommend its publication, for the main following reasons: Before proceeding to the discussion of the modifications in the manuscript we want to mention that it was substantially changed. Therefore, it will be impossible to describe the detail of every modification in this answer. We remit the reviewer to the annotated version of the manuscript that we are attaching together with the final revised one. In what follows we will discuss the spirit of the modifications made.

Comment 1: Once optimized, the model predictive skills are still not convincing, at least based on figure 4. Indeed, there might be an inconsistency between this figure, where the model departs from the observations by 0.5-1m a large part of the time, and figure 3 and 5, which suggests errors < 0.5 m. Considering that the correct figure is figure 4, I would rather expect ERMS of the order of $\sim$0.3-0.4 m, which is not convincing compared to recent studies (e.g. Bunya et al., 2010 ; Bertin et al., 2015). A possible reason for these relatively weak predictive skills is the wind that was selected (NCEP), which is way too coarse to compute storm surges accurately. From my own experience, a wind forcing of at least 0.2 resolution should be employed. Why not considering the CFSR reanalysis (Saha et al., 2010), which is fully available and offers a spatial

resolution of 0.2 and a time resolution of 1 h.

Following your advice in this and other comments, a number of changes were made in the model, simulations and forcings, so as in the analysis of the results: 1) Two different atmospheric data bases were used in order to compare results and to observe the improvements that result from a better resolution on the wind forcing (see section 3.4 and 4.1 of the revised manuscript). We used NCEP/NCAR-RI and ERA-INTERIM. 2) We incorporated the changes in the wind speed in the parameterization of the wind drag coefficient (Eq. 4, Section 3.4). 3) We increased the number of simulations in the Morris analysis from 200 to 300, what allows to make an statistic with 50 samples. 4) All the simulations and analysis were extended to one full month (May, 2000), including a large storm surge. Accordingly, the intervals of existence of the wind related parameters were recomputed. 5) For the analysis, we separated the "short time scales" (periods less than 30 hours) from the "long time scales" (periods more than 30 hours). The "long time scales" will mainly reflect the effect of the surge; the "short time scales" include the tides and the atmospheric processes related to sea breeze. Results show a general improvement for both wind databases (see for instance, figure 4 and table 3) in comparison with the original manuscript. A significant improvement is observed when wind resolution increases (from NCEP to ERA-INTERIM). For NCEP-NCAR to ERA-INTERIM, ERMS reduces from 0.15 to 0.12 at Palermo station and from 0.21 to 0.13 at Oyarvide. The performance of the model for ERA-INTERIM is satisfactory and comparable to other recent studies.

Comment 2: The authors evaluated directly total SSH, which does not allow to understand if their errors come from tides, storm surges or a bit of both. What is the impact of baroclinic circulation in such a large estuary, baroclinic effects are even not mentioned in the paper? I would suggest separating model results for tide only, surge only and model +surge. This would namely improve understanding, tide-surge interactions (e.g. Idier et al., 2012), which might be quite important in such a large estuary. I understand that it would lengthen the paper substantially but the present paper is quite

short compared to the average of NHESS papers. You are right in the sense that this had not been commented in the manuscritp. Therefore, we have several comments regarding yours: 1) The impact of the baroclinic circulation was not considered in the simulations of this manuscript for two main reasons: a) The upper and intermediate Río de la Plata, where our manuscript is mostly concentrated on, is filled with freshwater; the influence of oceanic water begins to be felt only offshore the Barra del Indio Shoal (see, for instance, Guerrero et al., 1997 for salinity field maps). b) In this work we are interested exclusively in the SSH, which is a barotropic process, not dependent upon the baroclinic structure (Gill, 1982). We included a comment about this in the manuscript (see Section 3.1). 2) Following your advice, for the analysis of the solutions, we divided them in "short time scales" and "long time scales" (see answer to the previous comment for the definition of those scales). It resulted that the model (with the introduced changes) has a better skill for the surge (or "long time scales") than for the "short time scales". Even though this can seem curious at the first glance, the fact is that in the Río de la Plata "short time scales" are not only related to tides but also to short timescale wind forced processes, as for instance, the sea breeze, (Simionato et al., 2005) that probably are not well represented in the reanalyses. We conclude, anyhow, that the model in its present form has a satisfactory skill for both time scales, with RMSE of the order of 0.1 m

3) Following your suggestions, we eliminated the former sections of nonlinear interactions between the wind speed and the runoff, and included:

a) A new section analysing the nonlinear interactions between the surge, the tide and the continental discharge (Section 4.2 of the revised manuscript); for this, simulations with all the forcings (tides, runoff and surge), only with tides, only with surge and only with continental discharge were run, analysed and discussed.

b) Another section was added quantifying the effects of changes in the runoff (Section 4.3 of the revised manuscript). The analysis you suggested clearly improved the article, as results are very interesting and novel for the Río de la Plata. It is found that

the nonlinear interactions are large in the Río de la Plata (accounting for approximately 10% of the surge), and that the inclusion of tides and runoff is fundamental in a forecast model for this estuary. In this sense, the "Results" section was widely changed and improved, including recommendations for the construction or the future forecast models as follows: "In this work, we discussed a sensitivity analysis (SA) based on Morris methodology, which is particularly well suited for models with large computational demand, to determine the sensitivity of numerical solutions for the Southwestern Atlantic Continental Shelf with emphasis in the wide and fast flowing RdP estuary to different parameters. An evaluation of the overall model SA the most critical storm event for the inhabitants of the region and for navigation, known as Sudestada, was permormed. The results from the SA reduce the required number of simulations needed for model calibration, reducing the future work to the fine calibration of the most sensitive inputs. ROMS_AGRIF model was chosen to build the pre-operational forecast model. It was applied in a hierarchy of 2-D one-way nested grids with refinement of the solutions over the RdP estuary. The SA was made including the bottom friction quadratic (cD) and linear (cl) parameterizations. Due to the scarcity of direct wind observations over the estuary and the limitations in the numerical modelling of the winds in the area, wind data becomes a significant source of errors and uncertainties for any ocean forecast model. Hence, wind speed (through a factor I) and direction ($\Theta$) were included in the SA. Finally, the RdP is very mighty, and continental discharge can vary significantly (in a range of around 80,000 m3 s-1) in the period of a few months, becoming also an important input which influence must be assessed. The ranges of existence of every input were set using values from literature, the RMSE with respect to observations, and extremes observed values, respectively. The sensitive analysis showed significant model response to all the considered inputs. The most important, with nonlinearity in the model response, was the wind speed (I). In particular, the model response showed to be very sensitive even to small changes in this forcing. The next most important input is Q, which response is more linear and presents a regional dependence, becoming less important towards the outer estuary (i.e., downstream). Finally, model solutions

are relatively much less sensitive to Θ, cD and cl. With the objective of further helping on the decision of how to built a numerical forecast strategy for SSH anomaly in the RdP, we also analyzed the interactions between the surge, the tide and the runoff. Results indicate that the interactions are important, accounting for around 10% of the total SSH anomaly during the storm. The most significant interaction (approximately 90% of the total) occurs between the surge and the tide, maximizing at Samborombón Bay and the upper RdP. The interaction between the tide and the runoff is much weaker, of the order of 10% of the amplitude of the tide. Finally, the interaction between of the runoff and the surge is of similar order of magnitude than that of the tide with the runoff. The last two interactions maximize at the upper estuary (where the tributaries flow to the RdP) and decay offshore, being almost negligible at the outer RdP. The results of this research provide information that will allow an optimal calibration of the model with only a fine tuning and a minimum number of simulations in the next future. They also highlight some the needs to face the construction of an accurate numerical forecast system for the prediction of extreme surges in the RdP. In this sense, we can conclude that: a) The fact that the model solutions are extremely sensitive to small uncertainties in the wind speed indicates that the most obvious way of improving the surge forecast is either improving the atmospheric forcing or at least quantifying the forecast error due to the uncertainties. Some ways of improving the wind forcing is by increasing the temporal and spatial resolution, and the diversity of physical processes included in the simulations, by the use of regional numerical models and/or assimilating data on the simulations. For this, more direct observations over the RdP would be necessary. As an intermediate step, an empirical adjustment of the winds could be attempted. The uncertainties in the SSH anomaly forecast can be quantified by ensemble modelling.

b) The inclusion of the continental discharge in a forecast model for the SSH anomaly in the RdP is fundamental. Its main effect is to introduce a setup (or SSH elevation), but also interacts with the tide and the surge, particularly in the upper estuary, where the most populated areas of the RdP coasts are located and where, in consequence, the impact of the floods maximizes. Nevertheless, the fact that the variability of the runoff

is uncoupled with the surge, warranties that small uncertainties in the value of the discharge will not introduce large errors in the surge forecast. In this sense, for short term forecast the coupling of a hydrological model to the hydrodynamic one is unnecessary. c) Finally, it is absolutely necessary to include tides in the simulation. The tide has strong interactions with the surge, accounting for approximately 10% of the total signal. Furthermore, the tide interacts with the runoff, introducing more modifications in the real surge."

Finally, the title of the manuscript was changed to better reflect the new contents to " Model sensitivity and nonlinear interactions during extreme sea level events in a wide and fast-flowing estuary: the case of the Río de la Plata". With the new sections the revised version of the manuscript is considerable longer, including 9 figures and 3 tables.

Comment 3: There is a confusion between wind speed and surface stress. Why evaluating surface stress and not wind speed directly? Indeed, using the surface stress implies defining a drag coefficient, which is a very large source of uncertainty, namely because it depends on wind speed but also, for a given wind speed, on the sea-state. In the end of section 2.4, the author state that "the wind drag coefficient has a relative small effect on the parameterization. . .": what does it means given that Cd increases linearly with the wind speed? What is the idea behind keeping Cd constant while it is known that it increases with wind speed? Considering the sensitivity analysis on the wind speed where I ranges from 0.5 to 0.5, eq. (3) implies that Cd should vary by a factor of 3. We already mentioned (see answer to comment 1) we now evaluate directly wind speed and included the wind speed changes in the parameterization of the drag coefficient.

Answer to minor comments:

-P2, L12: "surface stress" is more common. Done.

-P2, L18-23: strange to start the introduction by presenting the study area. The introduction was modified and divided into two parts. All the description of the Río de la Plata was moved to a new "Study area" section and now the introduction concentrates on the surges, the nonlinear interactions and the aims of the paper.

-P2, L25: what do you mean by "system", has ENSO broader impacts than river runoff? We are referring to the Río de la Plata basin (the Río de la Plata estuary and its tributaries). We changed "system" to "runoff" in the manuscript to make it clearer.

-P2, L35: "along its margins" Done

-P3, L5: I strongly doubt that the RdP is located in one of the most cyclogenetic region of the world. Regarding tropical storms, the Southern Atlantic is on the contrary the only Ocean where the cyclonic activity is almost nil (https://earthobservatory.nasa.gov/IOTD/view.php?id=7079). Regarding extratropical storms, please consider that, in winter, storms develop continuously at high latitudes. You are right in the sense that we did not make clear in the original manuscript that we were referring to the subtropics. The new version of the manuscript was changed in consequence and more references were added. The revised introduction starts as follows: The area between 30 °S and 45 °S is characterized by one of the highest cyclogenetic activities within the Southern Hemisphere (e.g., Necco, 1982; Sinclair, 1994; Gan and Rao, 1991). Over that area, cyclogenesis events have a mean frequency of around 120 events by year (Gan and Rao, 1991), with higher frequency during winter and spring. These events are associated to atmospheric waves that move along subtropical latitudes of the South Pacific and South American regions (Vera et al., 2002). When cyclones develop over Uruguay, they can yield very strong and persistent southeasterly winds, known as "Sudestadas", with speeds that can easily exceed 15 m s-1 (Seluchi, 1995; Seluchi and Saulo, 1996).

-P3, L8: "yields" rather than "originate". Done.

-P3, L16-20: please explicit the corresponding surges. The provided information refers to the maximum of the surge. It was clarified in the text.

-P3, L30: a forecast is never "exact", rather use "quality". Done.

-P4, L7: explain why a wide estuary turns more challenging the estimation of the wind speed and direction. The main difficulty is that in the Río de la Plata, small scale processes in the atmosphere have an impact on the water response. This is the case, for instance, with the sea breeze, that has been proved to account for a significant portion of the variance in the Río de la Plata currents (Simionato et al., 2005). Not all the atmospheric models include those processes, for what improving winds is not only a matter of resolution but also of including mor processes in the simulations. It was clarified in the revised manuscript (please see Section 4.1).

-P4, section 2.1: rather explain what are the key differences of ROMS_AGRIF compared to the original version. As said above, please better justify why baroclinic circulation can be neglected. The main reason to utilize this version of ROMS is that it has routines from MARS and HYCOM that we expect to apply in the future (for instance, sediments transport). It must be emphasized that the hydrodynamics (are part of the model we use in this work) are the same in both versions of ROMS. AGRIF model also includes the possibility of 2-way interaction, what can be useful in the future. The reasons why baroclinic circulation can be neglected in a surge model for the Río de la Plata was discussed above (please see answer to Comment 2).

-P5, L9: "aims to establish". Done

-P5, L30: this is not true that RMSE is "very much related" to the correlation. For instance, let's consider a perfect model (RMSE = 0 m) that you divide by a factor of 2: the correlation coefficient will be stick to 1 while the RMSE will get large. You are right. The sentence was erased.

-P6, L17: what is the reference height for this open water wind measurements? Usually, this is hardly 10 m and therefore wind speed cannot be compared directly with 10 m wind speed from atmospheric models. If a correction is made for the comparison for instance applying a logarithmic model, which Z0 is used? All the observations are

reported at a height of 10 m and therefore they are supposed to be comparable. It was clarified in the revised manuscript.

-P7, L9: rather "momentum balance". Done.

-P7, L12-18: why combining a linear with a quadratic bottom stress? This is not a common procedure. Please better justify through adequate references. Eq. 2 is the standard procedure used by ROMS_AGRIF. Most of the models ignore the linear bottom stress. In our case, we just decided to evaluate the impact of this parameter together with the more usual quadratic coefficient. It was found that actually simulations are almost unsensitive to cl.

-P7, section "c": if runoff is that important, why baroclinic circulation is not? Also, better justify through references why tide-surge interactions are huge? As above mentioned, the baroclinic processes (associated to the vertical density structure) do not significantly affect SSH, specially in the upper and intermediate Río de la Plata, filled by freshwater. Instead, the large runoff of the tributaries at the estuary head, both changes the mean sea level (Fig. 9) and introduces large currents which interact with the tides and the surge (see Fig. 8). We do not understand your last comment as we never mentioned the tide-surge interactions in the original manuscript.

-P8, eq. (3): where this parameterization comes from? Smith and Banke? Large and Pond? The corresponding reference was included. It comes from Bowden (1983).

-P10, L5: what does "it" stands for? This paragraph was erased.

References:

Raúl A. Guerrero, Eduardo M. Acha, Mariana B. Framiñan, Carlos A. Lasta, Physical oceanography of the Río de la Plata Estuary, Argentina, Continental Shelf Research, Volume 17, Issue 7, 1997, Pages 727-742, ISSN 0278-4343, http://dx.doi.org/10.1016/S0278-4343(96)00061-1.

Gill, A. (1982) Atmosphere-Ocean Dynamics. Academic Press, New York.

Answer to Reviewer 2 comments:

As we mentioned to the Editor, we are grateful for your revision which certainly contributed to a significant improvement of the article. Your comments were carefully considered and changes were done accordingly.

This is a manuscript which fits the theme of NHESS but which needs major revisions before publication. Before proceeding to the discussion of the modifications in the manuscript we want to mention that it was substantially changed. Therefore, it will be impossible to describe the detail of every modification in this answer. We remit the reviewer to the annotated version of the manuscript that we are sending together with the final revised one. In what follows we will discuss the essentials of the modifications made in relation to your comments.

The writing can be improved. Some sections can be shortened or summarized. The manuscript was almost fully rewritten. Sections were shortened and summarized as much as possible, and a full revision of English usage was made. In particular: 1) The original "Introduction" section was much reduced, and a discussion on nonlinear interactions between the surge, the tide and the runoff in estuaries was included. Efforts were made to clearly highlight the aims of the work.

2) A new "Study area" section was added, where the Río de la Plata Estuary and its impact is discussed.

3) Morris analysis is now presented for two different wind data bases of the different resolution and the differences in the model response are discussed.

4) The former Section 3.2 was eliminated.

5) A new complete section discussing the nonlinear interactions between the tide, the surge and the runoff was included.

6) In a last section the effect of changes in the runoff is quantified.

7) The "Conclusions" section was fully rewritten, as follows:

"In this work, we discussed a sensitivity analysis (SA) based on Morris methodology, which is particularly well suited for models with large computational demand, to determine the sensitivity of numerical solutions for the Southwestern Atlantic Continental Shelf with emphasis in the wide and fast flowing RdP estuary to different parameters. An evaluation of the overall model SA the most critical storm event for the inhabitants of the region and for navigation, known as Sudestada, was permormed. The results from the SA reduce the required number of simulations needed for model calibration, reducing the future work to the fine calibration of the most sensitive inputs. ROMS_AGRIF model was chosen to build the pre-operational forecast model. It was applied in a hierarchy of 2-D one-way nested grids with refinement of the solutions over the RdP estuary. The SA was made including the bottom friction quadratic (cD) and linear (cl) parameterizations. Due to the scarcity of direct wind observations over the estuary and the limitations in the numerical modelling of the winds in the area, wind data becomes a significant source of errors and uncertainties for any ocean forecast model. Hence, wind speed (through a factor I) and direction ($\Theta$) were included in the SA. Finally, the RdP is very mighty, and continental discharge can vary significantly (in a range of around 80,000 m3 s-1) in the period of a few months, becoming also an important input which influence must be assessed. The ranges of existence of every input were set using values from literature, the RMSE with respect to observations, and extremes observed values, respectively. The sensitive analysis showed significant model response to all the considered inputs. The most important, with nonlinearity in the model response, was the wind speed (I). In particular, the model response showed to be very sensitive even to small changes in this forcing. The next most important input is Q, which response is more linear and presents a regional dependence, becoming less important towards the outer estuary (i.e., downstream). Finally, model solutions are relatively much less sensitive to $\Theta$, cD and cl. With the objective of further helping on the decision of how to built a numerical forecast strategy for SSH anomaly in the RdP, we also analyzed the interactions between the surge, the tide and the runoff. Results indicate that the interactions are important, accounting for around 10% of the total SSH anomaly during the storm. The most significant interaction (approximately 90% of the total) occurs between the surge and the tide, maximizing at Samborombón Bay and the upper RdP. The interaction between the tide and the runoff is much weaker, of the order of 10% of the amplitude of the tide. Finally, the interaction between of the runoff and the surge is of similar order of magnitude than that of the tide with the runoff. The last two interactions maximize at the upper estuary (where the tributaries flow to the RdP) and decay offshore, being almost negligible at the outer RdP. The results of this research provide information that will allow an optimal calibration of the model with only a fine tuning and a minimum number of simulations in the next future. They also highlight some the needs to face the construction of an accurate numerical forecast system for the prediction of extreme surges in the RdP. In this sense, we can conclude that: a) The fact that the model solutions are extremely sensitive to small uncertainties in the wind speed indicates that the most obvious way of improving the surge forecast is either improving the atmospheric forcing or at least quantifying the forecast error due to the uncertainties. Some ways of improving the wind forcing is by increasing the temporal and spatial resolution, and the diversity of physical processes included in the simulations, by the use of regional numerical models and/or assimilating data on the simulations. For this, more direct observations over the RdP would be necessary. As an intermediate step, an empirical adjustment of the winds could be attempted. The uncertainties in the SSH anomaly forecast can be quantified by ensemble modelling.

b) The inclusion of the continental discharge in a forecast model for the SSH anomaly in the RdP is fundamental. Its main effect is to introduce a setup (or SSH elevation), but also interacts with the tide and the surge, particularly in the upper estuary, where the most populated areas of the RdP coasts are located and where, in consequence, the impact of the floods maximizes. Nevertheless, the fact that the variability of the runoff is uncoupled with the surge, warranties that small uncertainties in the value of the discharge will not introduce large errors in the surge forecast. In this sense, for short term forecast the coupling of a hydrological model to the hydrodynamic one is unnecessary.

c) Finally, it is absolutely necessary to include tides in the simulation. The tide has strong interactions with the surge, accounting for approximately 10% of the total signal. Furthermore, the tide interacts with the runoff, introducing more modifications in the real surge."

Finally, the title of the manuscript was changed to better reflect the new contents to " Model sensitivity and nonlinear interactions during extreme sea level events in a wide and fast-flowing estuary: the case of the Río de la Plata".

Sometimes (too) many references are attributed to a simple sentence. An example, from the Introduction: "The RdP has a huge runoff with a mean value of around 22,000 m3 s-1, ranking 5th worldwide in water discharge (Nagy et al., 1997; Jaime et al., 2002, Framiñan et al., 1999)." I wouldn't use the terms "huge", nor "runoff". Also don't see the relevance of worldwide ranking. Also don't see why such a minor sentence deserved 4 citations. It was corrected and references were reduced to those necessary.

The authors reach a somewhat trivial set of conclusions, in that *storm* surge results are most sensitive to the wind forcing and to the bottom roughness – what else should be expected in such a shallow, wide estuary? We apologize, as after rereading the manuscript we understand that the conclusions were poorly written in the original manuscript. The new version of the paper was essentially revised and new studies were added. The following major changes were made: 1) Two different atmospheric data bases were used in order to compare results and to provide an idea of the improvements that would result from a better resolution on the wind forcing (see section 3.4 and 4.1 of the revised manuscript). 2) For the analysis, we separated the "short time scales" (periods less than 30 hours) from the "long time scales" (periods more than 30 hours). The "long time scales" will reflect the effect of the surge; the "short time scales" include the tides and the atmospheric processes related to sea breeze. 3) A new section analysing the nonlinear interactions between the surge, the tide and the continental discharge was included (Section 4.2 of the revised manuscript); for this, simulations with all the forcings (tides, runoff and surge), only with tides, only with

surge and only with continental discharge were run, analysed and discussed. 4) Another section was added quantifying the effects of changes in the runoff (Section 4.3 of the revised manuscript).

The new analyses improved the scope and conclusions of the article, as results are interesting and novel for the Río de la Plata and highlight the needs of a future forecast model for the Río de la Plata.

A few other remarks: - Would improve if one or more figures showed the model grids used; As the model grids are regular, plotting them does not provide much useful information to the reader. We tried to built such a figure, but it resulted too dirty.

- Why use an ocean model to simulate an estuary of 10m average depth? More flexibility would yield finer resolutions; We decided to use an oceanic model because we are also interested in the adjacent continental shelf and in the future we expect to include the baroclinic processes, so as the sediments transport. The chosen model will also allow the study of 2-way interactions.

- The discussion & conclusions is missing a more comprehensive comparison against similarly-minded papers, e.g. Zijl et al. (2015) where RMSE's are much smaller; The aim of our manuscript is on the sensitivity of the solutions to the diverse inputs and on the nonlinear interactions between the tide, the surge and the runoff. RMSE was primarily used in the Morris analysis as the chosen output. With regards to the RMSE as a measure of model skill, a number of improvements were introduced in the simulations and forcings. The performance of the model forced with ERA-INTERIM winds is satisfactory and comparable to other recent studies.

- The discussion & conclusions would benefit from a clear separation of tidal from other mechanisms contributing to the total water levels. Following your advice and that of Reviewer 1, for the analysis of the solutions, we divided them in "short time scales" and "long time scales" (see answer to the previous comment for the definition of those scales). It resulted that the model (with the introduced changes) has a better

skill for the surge (or "long time scales") than for the "short time scales". Even though this can seem curious, the fact is that in the Río de la Plata "short time scales" are not only related to tides but also to short timescale wind forced processes (as for instance, the sea breeze, Simionato et al., 2005) that probably are not well represented in the reanalyses. We conclude, anyhow, that the model in its present form has a satisfactory skill for both time scales.

Please also note the supplement to this comment:
https://www.nat-hazards-earth-syst-sci-discuss.net/nhess-2016-393/nhess-2016-393-AC1-supplement.zip